# Effects on the Properties of Self-Compacting Cement Paste (PAA) with the Addition of Superabsorbent Polymer

**DOI:** 10.3390/ma15238478

**Published:** 2022-11-28

**Authors:** Michel Henry Bacelar de Souza, Bárbara Almeida Teixeira, Paulo Cesar Gonçalves, Lucas Ramon Roque da Silva, Míriam de Lourdes Noronha Motta Melo, Vander Alkmin dos Santos Ribeiro, Daniela Sachs, Patrícia Capellato, Valquíria Claret dos Santos

**Affiliations:** 1Institute of Natural Resources, Unifei-Federal University of Itajubá, Av. BPS, 1303, Itajubá Cep 37500 903, MG, Brazil; 2Institute of Mechanical Engineering, Unifei-Federal University of Itajubá, Av. BPS, 1303, Itajubá Cep 37500 903, MG, Brazil; 3Centre for Studies and Innovation in Biofunctional Advanced Materials, Institute of Physics and Chemistry, Unifei-Federal University of Itajubá, Av. BPS, 1303, Itajubá Cep 37500 903, MG, Brazil

**Keywords:** self-compacting paste (PAA), superabsorbent polymer (SAP), internal cure

## Abstract

The addition of Superabsorbent Polymer (SAP) decreases the effect of autogenous shrinkage present in pastes, mortars, and concretes. In this study we investigated the influence of the addition of SAP in self-compacting cement paste mixtures. Eighteen 5 × 10 cylindrical specimens were molded in all, three for each mixture (CPII base, CPII 0.15%SAP/600μm, CPII 0.15%SAP/800 μm, CPV base, CPV 0.15%SAP/600 μm, CPV 0.15%SAP/800 μm). Two types of cement were tested, CP II-Z and CP V-ARI with 0.15% of weight replaced per two diameters of SAP (600 μm and 800 μm). The samples followed the standards required. Mini slump tests were carried out in the fresh state, and uniaxial compressive strength, elastic modulus, specific mass, absorption, and air content in the hardened state after 28 days. The results obtained show the SAP is high indicated to replaced cement in small % of weight i/to fresh and hardened paste. Likewise, the group mix n° 3 composed of CPII 0.15% of SAP with 800 μm diameter presented the best result.

## 1. Introduction

The construction industry’s technology has been updating and implementing new concepts, such as the usage of numerous alternative material such as superabsorbent polymer (SAP), in pastes, concretes, and mortars. These polymers, also known as hydrogels, are cross-linked polymers that have the ability to absorb a large amount of liquid, expanding to form an insoluble gel.

According to Souza et al. [1] the number of articles published in the area of concrete/mortar/paste and superabsorbent polymer throughout history showed an increase in the number of publications in the two databases analyzed (Scopus and Web of Science (WOS)). The authors showed a promising scenario for publications related to this subject, especially because the last 5 years accounted for more than 70% of all publications throughout history.

According to Gomes and Barros [2], self-compacting concrete is characterized by its ability to be compacted in the molds only by the action of its own weight, with no need to use vibration equipment. The most specific and important properties of this type of concrete are workability and stability, provided by specific additions of materials, such as superplasticizer and pozzolanic materials [3]. Concrete is self-compacting if it achieves the necessary fluidity and cohesion and resists segregation. To ensure that these properties are achieved, tests are performed in the fresh state such as the slump test and the V funnel, where the values obtained should be consistent with those found in the literature [3]. Driven by the development of the construction industry in the search for special concretes, these properties motivated the research on the use of SCC, considered a promising advancement of conventional concrete.

Initially, the great interest in the use of SAP was in the area of polymer chemistry capable of absorbing large volumes of water. Known as “superabsorbents”, they have been used mainly for agricultural and forestry soil preparation [4]. SAP is generally obtained by the polymerization of acrylic acid, acrylic ethers, acrylamide, and other unsaturated monomers, being able to absorb from 100 to 1000 g of water per gram of dry SAP; such a process is accomplished by accommodation between the molecules [4].

Recently, the usage of superabsorbent polymers has sparked attention in civil engineering, notably in the manufacture of concrete. Research on the production of concrete with the addition of SAP has primarily focused on its use as an internal curing agent, because it has been demonstrated to be able to release water in a controlled amount according to the demand of the concrete during the curing process, thereby mitigating shrinkage during hydration reactions in low weight/mass mixtures [5].

The curing process is very important for the production of cement paste, however, during this process a very common problem occurs in works with large volumes of concrete: autogenous shrinkage. With it, cracks appear in building elements (walls, pillars, and others) which can compromise the structural integrity of the work by reducing its service life. Paying attention to this fact, the study of SAP incorporation in pastes, mortars and concretes is very promising because it can mitigate this process, helping in the curing process of the mixture. Autogenous shrinkage is a phenomenon that occurs in high-performance concretes because they have a very low water/cement ratio, which reduces the amount of water needed for cement-hydration reactions [6].

Studies have shown that SAP can significantly influence the amount of superplasticizer used in the mixture, increasing its consumption. Moreover, depending on the curing method adopted, the polymer can improve the hydration reactions and increase the mixture strength, or it can induce voids in the matrix and impair its performance in compression tests. Finally, a relationship was also observed between the amount of SAP added and performance in mechanical compression tests of the mixtures, where it was found that a higher addition rate of the polymer can lead to a decrease in mechanical strength in the tests of specimens [7].

Many studies on the impact of SAP on conventional concretes have been published, but there is still a significant demand for investigations on the behavior of SAP in self-compacting concretes.

In order to achieve the required fluidity and high strength concrete, a low water-to-cementitious ratio is essential for self-compacting cementitious materials mixture. The purpose of internal curing is to prevent autogenous stresses and deformations, which can lead to early cracking, by maintaining saturated conditions within a hydrated cement paste [6].

Tests with self-compacting high performance concrete, using the cement CEM II/A-D 52.5R, were performed for two mixtures, both containing polypropylene fibers, and one of them with the addition of SAP. The samples were submitted to a high temperature, around 1000 °C, for 90 min, and a visual analysis showed that the concrete with SAP did not suffer fragmentation, unlike the reference concrete. It is assumed that the mechanisms in which the SAP acts when exposed to high temperatures unite the polypropylene fibers, allowing their percolation at a low volume content, around 0.2% of the volume of concrete [8].

Al-Hubboubi et al. [6] studied the influence of SAP on the mechanical properties of concrete, whose experimental program involved four different mixtures; the reference mixture, two mixtures with partial replacement of sand by crushed brick waste, and one mixture with addition of 0.5% by weight of cement by SAP. The results of the axial compression tests showed an increase in strength from the age of 49 days, reaching approximately 70 MPa at 91 days. The results of the tensile tests showed that the concrete produced with the addition of SAP showed an increase in strength of around 40% compared to the reference mix. Another interesting analysis developed by the authors refers to the behavior of concrete with the addition of 0.5% of SAP produced with ambient curing and temperature elevation in relation to the control temperature. The results of compression tests showed an increase in strength in the order of 14% compared to the compressive strength of the reference mixture, showing the influence of SAP in the internal curing process of concrete.

In the scope of self-compacting cement, Baloch et al. [9] presented the study of self-compacting pastes (SCC) produced with the addition of SAP, whose mixtures incorporated nano silica (NS), super plasticizer (SP), and two of them with the addition of SAP. In the studies of the shrinkage of specimens as a function of time, the authors noted that the specimens with incorporation of NS and addition of SAP presented shrinkage considerably higher than those obtained with reference specimens, on the order of 3.33% for the same time interval. Furthermore, according to the analyses of BALOCH et al. [9], the samples cured in water presented pore formation in the matrix by the addition of SAP, however, in the samples cured in the open air, there was an improvement in the mechanical resistance, a fact that can be attributed to the capacity of internal cure of SA, improving the microstructure. The authors concluded that the addition of SAP allowed mitigating shrinkage as an internal curing agent, both with self-cured pastes (mainly, the air-cured samples), and with conventional pastes with or without the addition of nano silica particles. The results revealed that even with the addition of water to the composition, SAP increased the need for SP in the manufacture of PAAs, and its hydrogel action may be responsible for the decrease in flow. The effect of NS on pastes, on the other hand, was more obvious in the Flow test compared to SAP, since it significantly raised the demand for SP.

Chindasipiphan et al. [10] developed an experimental program with few conventional concrete mixtures with cement replacement by different proportions of fly ash and with addition of SAP. The specimens whose traces were composed with replacement of cement by 45% of fly ash and with addition of SAP in the proportions of 4.0%, 6.0% and 8.0% showed reduction of compressive strength by approximately 28%, 36% and 34%, respectively, in relation to the control trait. However, by means of a self-healing efficiency test, which was evaluated by temporal decreases in water discharge through a pre-existing crack, 100% closure of the crack and restoration of permeability at 28 days of healing were observed, indicating the medium-term healing capacity of concrete by the coupling effect of fly ash and SAP. In addition, Chindasipiphan et al. [10] were successful in using SAP as a crack sealer because it accelerates carbonation activities by depositing calcium carbonate in the cracks.

Another interesting alternative for the use of SAP in cementitious materials is in the control of shrinkage in self-leveling cementitious mortar. YANG et al. [11] evaluated the effect and mechanism of shrinkage compensation of self-leveling mortar by means of specimens with addition of SAP in the proportions of 0.2%, 0.4%, and 0.6% by weight. The test results of these specimens were compared with the reference sample, conventional mortar, and showed that internal curing using SAP had a satisfactory effect on drying shrinkage and self-leveling shrinkage compensation that decreased by 10% and 10.2%, respectively, for additions in the proportion of 0.4%.

The study of the microstructure of concrete samples with addition of SAP is a very interesting way to identify the influence of the superabsorbent polymer in the curing process. Through X-ray diffraction (XRD), thermogravimetric analysis (TGA), scanning electron microscopy (SEM), mercury intrusion porosimetry (MIP), and compressive strength tests, a concrete sample with 0.3% of SAP incorporation was evaluated. The incorporation of SAP resulted in a refinement of the porous structure of the paste, despite increasing the total porosity. In addition, the paste containing 0.3% SAP resulted in an intermediate calcium hydroxide content compared to the reference pastes, thus demonstrating that internal SAP curing water participates in the hydration reactions of the cementitious material [7].

This study aims to explore the use of alternative resources in the production of self-compacting cementitious materials by analyzing the mechanical and physical properties of pastes with the addition of SAP (in the fresh and hardened state).

## 2. Materials and Methods

### 2.1. Material Characterization

The aim of this study is to analyze the behavior of self-compacting paste under the addition of superabsorbent polymer (SAP). Therefore, the cement used must have a low heat of hydration, because a high heat of hydration would cause shrinkage (the hydration reactions are exothermic, i.e., release heat, and when this heat is too high, the mixture contracts, causing the phenomenon known as autogenous shrinkage, harmful to the mechanical properties of the paste). For this reason, two different types of cement were used: CPII-F and CPV-ARI, one with low and the other with high heat of hydration. Characterization tests of CPII-F and CPV-ARI were performed, being specific mass according to (ABNT NBR 16605:2017) [12] and laser granulometry, which still has no specific standard.

To improve the self-compactability of the paste, silica fume was added, a pozzolan (inorganic material that in the presence of water and calcium hydroxide works as a binder, increasing the cohesion of the particles of the mixture obtained as a by-product in the production of silicon metal and iron-silicon alloys) [13]. For the silica fume, specific mass tests were also performed according to ABNT NBR 16605:2017 [12] and laser granulometry.

The use of pozzolans, however, influences the water demand in the mixture, which has a direct effect on the mechanical properties of the material. In this study we chose to use a type II superplasticizer (whose dosage, in weight of cement, was provided by the manufacturer) because it significantly reduces the use of water without modifying the consistency of the mixture; such an additive aims to increase the workability of concrete, obtaining high values in the slump test and a good fluidity of the paste [14].

The business IGTPAN provided the requirements for the particular mass and absorption of the superabsorbent polymer that were required for this investigation.

### 2.2. Mortar Dosing

In general, there are several dosage methods for the production of concrete, mortar, and, in the case of this test, self-compacting cement paste specimens. We chose to use the values used in the Silva [15] dosage, which followed an essentially empirical scheme: besides data collected in other literature works, the method proposed by Tutikian [16].

Additions were made in place of cement, at a content of 0.15% of its weight, for two different maximum diameters: 600 μm and 800 μm, also varying the type of cement used: CP II-F and CP V-ARI.

The mixing process adopted consisted of mixing the dry components first, and then the incorporation of the liquid components gradually, as shown in Figure 1.

### 2.3. Properties of the Pulp in the Fresh State

To ensure the self-compacting properties, the slump test was used, which allows observing if the mixture presents fluidity and cohesion. The paste should present values within the limit of the data presented in the literature [2]. Figure 2 shows the apparatus and characteristics of this test.

The time it took the mixture to reach 115 mm (T115, in seconds), after the paste-containing container had been raised, should then be noted.

Curing requires regulating the temperature and flow of moisture to enhance the hydration of the cement. As the strength of the concrete section increases, it is vital to keep the temperature constant in order to prevent cracks produced by the strength diminishing [17].

To ensure the development of concrete’s physical attributes, such as mechanical strength and durability shown resistance, cool the material to temperature and humidity, particularly in the state and early ages. The curing process’ purpose is to enhance cement hydration and decrease shrinkage.

The continuity of curing is critical, and it must begin soon after molding. Curing late does not compensate for the lack of curing in the early moments after molding, committing the concrete to a limited performance. Proper curing may significantly minimize concrete shrinkage, and the consequences of not curing can be disastrous [18].

After the tests in the fresh state, the molded specimens remained in submerged curing in water solution with calcium hydroxide for 28 days.

### 2.4. Properties of the Paste in the Hardened State

With the hardened paste, experiments must be performed to verify its mechanical properties. In the case of this work, the following tests were performed: uniaxial compressive strength, dynamic modulus of elasticity, specific mass, absorption by immersion, and voids index.

The compressive strength test basically consists of applying successive force values, at a constant speed, to the specimen until its rupture [19]. The test consists of impacting the specimen surface, which is fixed on top of test equipment. The impact signal, in terms of frequency response, are received by data acquisition system [20]. The tests for specific mass, absorption by immersion, and void index are all governed by the same standard [21]. In the laboratory, it is necessary to weigh a specimen after drying in an oven (m_s_), after immersion in water (mi), and after drying the specimen with the help of a wet cloth (m_sat_). Mathematical relations are used to obtain the results of each property cited above, as shown in Table 1.

In this work, 18 cylindrical specimens of 5 cm in diameter and 10 cm in height were molded, as shown in Table 2, with the composition of the specimens for the aforementioned tests: uniaxial compressive strength, dynamic modulus of elasticity, specific mass, absorption by immersion and voids index. After deforming cylindrical specimens, curing took place in a solution of water and calcium hydroxide for 28 days. The following tests were performed: specific mass, absorption by immersion, voids index, dynamic modulus of elasticity and, finally, the destructive uniaxial compressive strength test with the same three specimens of each composition.

## 3. Results

### 3.1. Material characterization

The characterization results for specific mass according to (ABNT NBR 16605:2017) [12] and average particle diameter of the two cements and the silica fume are shown in Table 3.

In the case of superplasticizer, the SILICON AD 50404 was used, produced and manufactured by Tecnosil, where the recommended dosage is around 0.2 to 2.0% by weight of the cement. The superplasticizer SILICON AD 50404 has a specific mass of 1.1 (g/cm^3^) and absorption of 200 to 400 (g of water/g of SAP). For the superabsorbent polymer, the specifications are specific mass (g/cm^3^) = 1.1 and absorption (g of water/g of SAP) = 200–400 according to IGTPAN, 2022 [22].

### 3.2. Mortar Dosing

Table 4 shown traces of the reference mixture, with SAP added at 0.15%, and with diameters of 600 m and 800 m.

Table 4 shows that the mixtures with added CPV required more SP and a/c factor compared to CPII mixtures. The results in the fresh state went through the mini slump test, whose results are shown in Table 5, along with the limit values for the variables measured [4].

All pastes obtained a T115 suitable for the limit values of the results, which were obtained Gomes and Barros [2]. The D_f_ of the CPV paste mixture that contains 0.15% addition of SAP with diameter of 600 μm was the only mixture that did not reach the value of 180 ± 10 mm, not being considered self-compacting.

Table 5 presents the unit mixtures in mass of the tested pastes. According to Chindasipiphan et al. [10], the inclusion of SAP increased the need for superplasticizer. As can be seen, combination 1 consumes less superplasticizer and water relative to cement than the other CPII mixes, which might account for its poor performance in the fresh paste test.

The same happens for the consumption of water in mixture 5, which is lower than mixture 6, which could explain the D_f_ obtained, considering that both contain SAP in their compositions.

### 3.3. Mechanical Properties

The average results for compressive strength (MPa) are shown in Figure 3.

It is observed that the values of compressive strength for the mortars with CPII smaller diameter of 600 μm SAP there was a decrease in resistance of 13% and for the 800 μm an increase of 35%. For the CPV cement it was the opposite; for the smallest diameter, there was an increase of 66% and for 800 μm a decrease of 35%. Analyzing Table 6, it can be seen that the results of modulus of elasticity had the same behavior only for the mortar with CPII and SAP diameter of 800 μm; there was a decrease in the value of modulus of elasticity of 19%, a behavior opposite to that of compressive strength.

The greatest decrease in the value of compressive strength was for mixture number 6: CPV—0.15%SAP—800 μm. A graph called the Pareto Graph was also prepared in Minitab software, version 17, which relates the variables inserted into the mixture, percentage of addition of SAP and variation of its granulometry, with the performance of the compressive strength of the mixture (Figure 4).

The Pareto Chart of the standardizer effect shows which variables taken into consideration in the dosage of the paste are significant in modifying the measured responses. Variable A represents the percentage of SAP incorporated in the paste, B represents the diameters used for SAP, and AB represents the interaction between the two variables tested. In this case, according to the statistical parameters adopted, the variable that crosses the value of 2.306 is considered statistically significant. In particular the variables AB and B have a higher degree of significance process.

Analyzing Figure 4, it is possible to see that factor B–D_max_ of SAP—had the greatest influence on the results of compressive strength, thus, it is possible to attribute the poor performance of the strength of paste number 6 to the D_max_ of SAP equal to 800 μm used in the mixture. Such influence is also observable in mix number 5: CPV—0.15%SAP—600 μm, which showed better performance in the compressive strength test. Table 6 shows the results of Modulus of Elasticity (ME), Specific Mass (SM), Absorption (Abs) and Air Content (AirC) for all mixtures studied and average (μ).

The higher values of absorption and air content according to Table 6 for the mortars with CPII/SAP 600 μm and CPV/SAP 800 μm collaborating with the two lower values of compressive strength when compared to their base mixtures. Schröfl et al. (2012) [23] also verified that SAP particles in cementitious mixtures is created in the matrix a number of voids that can affect the strength of the material negatively. Agostinho et al. (2012) [24] studied that in CPV-ARI pastes with addition of 0.15% and 0.30% of SAP, there is a change in the hydration kinetics of cement because they showed peaks of hydration of C_3_S and lower rates of acceleration of reactions due to the occurrence of internal curing or the presence of extra water that changed the w/c ratio. The same fact may have contributed to the resistance gain in the samples CPII/SAP 800 μm and CPV/SAP 600 μm, remembering that CPII gained more resistance with the addition of SAP with larger granulometry and CPV had an increase in resistance with the addition of SAP with smaller granulometry.

However, Baloch et al. (2019) [9], state that the SAP can exhibit two behaviors: it will enhance hydration processes, assisting in the internal curing process and boosting strength, or it can induce voids in the matrix, weakening strength. This last theory explains the performance of mixture 6 in the axial compression test is that the inclusion of SAP in this mixture may have contributed to the second scenario reported, where the mechanical compression strength was decreased by creating voids in the cement matrix. In the study indicated above, there were two different methods of curing used: in water and dry. The mixes treated in water revealed matrix holes and had decreased strength, whereas the samples cured in air had increased strength.

Al-Habboubi et.al (2018) [6] studied the influence of SAP on the mechanical properties of concrete using a mixture of 0.5% by weight of cement by SAP. The results of the compression and tensile tests showed an increase in strength in both cases. When the concrete with SAP was produced in a hot climate, there was an increase of about 14% in compressive strength due to the internal curing method by SAP (concrete left in the air). Chindasipiphan et al. (2019) [10] performed tests with cement replacement by fly ash and with addition of 4.0%, 6.0%, and 8.0% of SAP the reductions in compressive strength were approximately 28%, 36%, and 34%.

The last two articles cited here did not bring information about the diameter of the SAP, but the percentage of SAP used in the mixture is an important factor in determining the strengths.

## 4. Conclusions

In the current study, the effects of using superabsorbent polymer (SAP) on the production of self-compacting cement paste in both their fresh and hardened states were examined. The main conclusions follow:The incorporation of SAP in cement pastes influenced the self-compacting properties due to the diameter of the SAP and the type of cement, whose slump tests showed greater inconsistency for the diameter of 600 μm.The mechanical properties were also influenced by the incorporation of SAP in the pastes tested. The results of the compressive strength test presented very interesting values, demonstrating an increase in strength in relation to the base mixture. The best performances were observed for the mixtures that had in their composition CPII cement with in-corporation of SAP of 600 μm diameter and for the 800 μm diameter, presenting an increase of 35.2% and 34.3%, respectively, in relation to the reference mixtures.The results of absorption and air content corroborate the analysis of the influence of the addition of SAP on the mechanical strengths. The CPV mixture with addition of SAP with maximum diameter showed results with the highest rates of absorption and air content, with an increase of 29.2% for the absorption and 33.6% for the content, compared with the reference mixture, substantiating the reduction in compressive strength of mixture 6.In general, cementitious paste mixtures with the addition of SAP improve the properties analyzed. However, due to the results found for mixture 6, future works can be studied for the dry curing processes.

## Figures and Tables

**Figure 1 materials-15-08478-f001:**
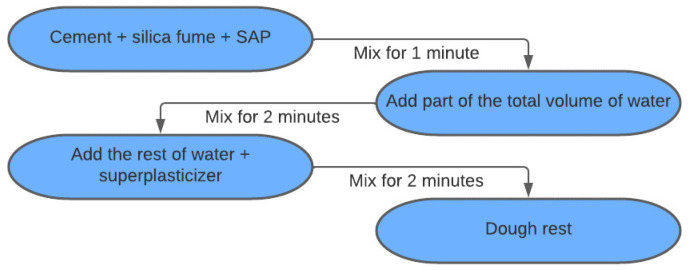
Schematic of the mixing procedure.

**Figure 2 materials-15-08478-f002:**
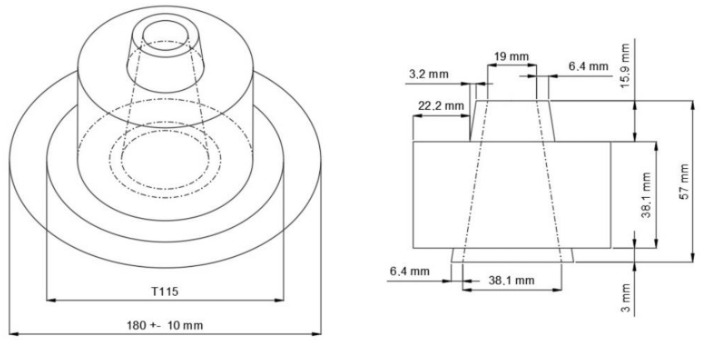
Equipment and illustration of the mini slump test for self-compacting paste. Source: [2].

**Figure 3 materials-15-08478-f003:**
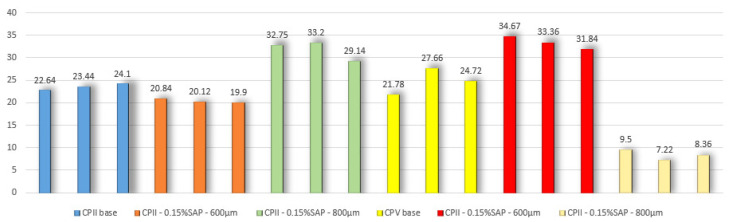
Compressive strength of CP’s at 28 days (MPa).

**Figure 4 materials-15-08478-f004:**
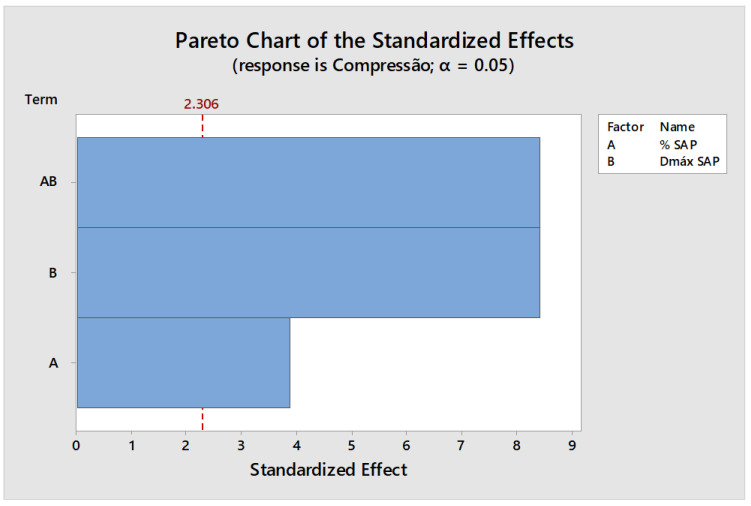
Effects of imposed factors on the performance of strength.

**Table 1 materials-15-08478-t001:** Mathematical formulations to obtain the properties.

Property	Mathematical Equation
Specific mass (ρ_s_)	ρ_s_ = m_s_/(m_sat_ − m_i_)
Absorption by immersion (A)	A = ((m_sat_ − m_s_)/m_s_) × 100
Void index (I_v_)	I_v_ = ((m_sat_ − m_s_)/(m_sat_ − m_i_)) × 100

**Table 2 materials-15-08478-t002:** Quantity and composition of the specimens.

Composition	CPII Ref.	CPII—0.15%SAP 600 μm	CPII—0.15%SAP 800 μm	CPV Base	CPV—0.15%SAP 600 μm	CPV—0.15%SAP 800 μm
Number of samples	3	3	3	3	3	3

**Table 3 materials-15-08478-t003:** Characterization test results of the cements used.

Cement Type	Specific Mass (g/cm^3^)	D50 (μm)
CPII-F 32	2.99	12.47
CPV-ARI	3.04	12.43
silica fume	2.20	39.78

**Table 4 materials-15-08478-t004:** Unit mass traces of PAA: reference and with added SAP.

N°	Mixture	Cimente	Silica	Superplasticizer (SP)	Water a/c	SAP
1	CPII reference	1	0.083	0.003	0.32	0.0000
2	CPII—0.15%SAP—600 μm	1	0.083	0.004	0.37	0.0015
3	CPII—0.15%SAP—800 μm	1	0.083	0.004	0.32	0.0015
4	CPV reference	1	0.083	0.004	0.34	0.0000
5	CPV—0.15%SAP—600 μm	1	0.083	0.005	0.36	0.0015
6	CPV—0.15%SAP—800 μm	1	0.083	0.005	0.39	0.0015

**Table 5 materials-15-08478-t005:** Results of the mini slump test.

N°	Mixture	T_115_ (s) Ref.	T_115_ (s)	D_f_ (mm) Ref.	D_f_ (mm)
1	CPII reference	Between 2 and 3.5	2.5	Between 170 and 190	180
2	CPII—0.15%SAP—600 μm		2.0		177
3	CPII—0.15%SAP—800 μm		2.5		170
4	CPV reference		2.5		170
5	CPV—0.15%SAP—600 μm		2.0		165
6	CPV—0.15%SAP—800 μm		3.0		171

**Table 6 materials-15-08478-t006:** Properties of the cement paste at 28 days.

Samples	Mixture	ME (GPA)		SM (g/cm^3^)		Abs (%)		AirC (%)	
		Unit.	μ	Unit.	μ	Unit.	μ	Unit.	μ
1	CPII base	14.9	15.38	2.3	2.28	12.2	11.77	13.9	13.35
		15.7		2.3		12.3		14.1	
		15.5		2.3		10.8		12.1	
2	CPII—0.15% SAP—600 μm	12.6	12.89	2.3	2.29	14.3	13.80	16.7	15.90
		13.7		2.3		13.6		15.7	
		12.4		2.3		13.5		15.3	
3	CPII—0.15% SAP—800 μm	12.4	12.43	2.2	2.21	11.2	11.05	12.6	12.42
		12.4		2.2		11.2		12.6	
		12.4		2.2		10.8		12.2	
1	CPV base	13.6	12.88	2.3	2.31	12.5	11.75	13.3	12.99
		12.5		2.3		12		13.6	
		12.6		2.3		10.8		12.1	
2	CPV—0.15% SAP—600 μm	12.8	13.94	2.4	2.24	11.5	10.95	13	12.22
		13.5		2.2		11.1		12.5	
		15.6		2.2		10.2		11.2	
3	CPV—0.15% SAP—800 μm	10.8	10.76	2.2	2.22	15.9	15.21	18.9	17.83
		10.3		2.2		14.3		16.3	
		11.1		2.2		15.5		18.3	

## Data Availability

Not applicable.

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
