# Peer review of "Effects on the Properties of Self-Compacting Cement Paste (PAA) with the Addition of Superabsorbent Polymer"

_materials, 2022, doi:10.3390/ma15238478_

Round 1

Reviewer 1 Report

This paper presents some results on effects of the addition of SAP on the SCC. The topic is interesting. There are some shortcomings which must be improved for being published.

1. The English including formats is badly written.

2. The logistics of this paper is chaotic, which must be rearranged.

3. The authors mention that SAP is critical to autogenous shrinkage of SCC, while there is no supportive data. I don't understand what is specific point of this paper.

Author Response

03th November 2022

The authors of the manuscript and I are very grateful for your review and considerations. Rest assured that all of your observations were carefully analyzed and they undoubtedly enriched the result of the work. Thus, you can find comments on the revisions made based on your review following.

Comments and Suggestions for Authors

Response to reviewer

This paper presents some results on effects of the addition of SAP on the SCC. The topic is interesting. There are some shortcomings which must be improved for being published.

  1. The English including formats is badly written.

Reviewers are right, changes have been made to all text.

  1. The logistics of this paper is chaotic, which must be rearranged.

It was verified so that the sequence presented in the methodology is the same in the results, and some sentences were rearranged.

  1. The authors mention that SAP is critical to autogenous shrinkage of SCC, while there is no supportive data. I don't understand what is specific point of this paper.

Reviewers are right, we changed the objective to be clearer.

Again, I appreciate your availability for reviewing and indicating improvements in the work. If you have any questions or other suggestions, please do not hesitate to contact me.

Sincerely,

Patricia Capellato

Reviewer 2 Report

The manuscript entitled " Effects of the addition of superabsorbent polymer on the production and curing of self-compacting cement paste (PAA)” is interesting, well-prepared, and totally within the scope of the journal. The authors investigated the influence of polymer in the production and curing of self-compounding cement paste. I recommend the Editor accept this manuscript after a minor revision.

I have some issues that must be answered before any decision.

1.     I recommend the authors include the contents related to the cost-effectiveness of the SAP concrete composites in the introduction part.

2.     What will be the interaction between SAP and iron rods in the concrete, as it has a high-water absorption ability will it increase the rusting of iron rods?

3.     What is the effect of temperature on SAP, can you explain its absorption ability in a wide range of temperature conditions? Is this concrete usable under all climatic conditions or have any limitations? 

Author Response

03th November 2022

The authors of the manuscript and I are very grateful for your review and considerations. Rest assured that all of your observations were carefully analyzed and they undoubtedly enriched the result of the work. Thus, you can find comments on the revisions made based on your review following.

Comments and Suggestions for Authors

Response to reviewer

The manuscript entitled " Effects of the addition of superabsorbent polymer on the production and curing of self-compacting cement paste (PAA)” is interesting, well-prepared, and totally within the scope of the journal. The authors investigated the influence of polymer in the production and curing of self-compounding cement paste. I recommend the Editor accept this manuscript after a minor revision.

I have some issues that must be answered before any decision.

  1. I recommend the authors include the contents related to the cost-effectiveness of the SAP concrete composites in the introduction part.

Unfortunately we could not find this information about the cost-effectiveness of the SAP concrete composites.

  1. What will be the interaction between SAP and iron rods in the concrete, as it has a high-water absorption ability will it increase the rusting of iron rods?

High water absorption is usually observed in the curing process.

  1. What is the effect of temperature on SAP, can you explain its absorption ability in a wide range of temperature conditions? Is this concrete usable under all climatic conditions or have any limitations? 

Since its qualities aid in the curing process, SAP shows to be the cement matrix's buddy in bad weather. SAP in its saturated form offers an additional volume of water, which is necessary in these conditions, in locations where water evaporation is increased owing to high temperatures, hence reducing the likelihood of shrinkage.

Again, I appreciate your availability for reviewing and indicating improvements in the work. If you have any questions or other suggestions, please do not hesitate to contact me.

Sincerely,

Patricia Capellato

Reviewer 3 Report

1.     The English language should be improved in the entire manuscript. The author may not translate fully from the Brazilian report into an English manuscript. In general, the English and paper format is quite poor, for example, from line 235 to line 240, Table 5.

2.     What are the significant findings of the manuscript? The author’s team should clarify the featured results of this research.

3.     The content of the Introduction should be revised into a condensed version. Some information can be shifted to the following parts.

4.     The research objectives and the significance of the research should be clarified in the manuscript.

5.     There is a lack of citations for the reviews and the literature content should be improved.

6.     The material properties used in this research should be clarified, especially the cement properties, and pozzolanic properties.

7.     The curing process of the specimens is not specified. The replicates for engineering tests were not shown.

8.     How to determine the 600μm and 800μm particle sizes?

9.     Why did the author’s team conduct the mini-slump test instead of the conventional slump test? It should be specified. The slump test results should be presented in Figure format.

10.  The unit of the unconfined compressive strength test was not shown.

11.  The author’s team should clarify why the mixture having CPV0.15% 800μm dropped significantly. Meanwhile, with the same cement type, the 600μm mixture obtained the highest compressive strength.

Author Response

03th November 2022

The authors of the manuscript and I are very grateful for your review and considerations. Rest assured that all of your observations were carefully analyzed and they undoubtedly enriched the result of the work. Thus, you can find comments on the revisions made based on your review following.

Comments and Suggestions for Authors

Response to reviewer

  1. The English language should be improved in the entire manuscript. The author may not translate fully from the Brazilian report into an English manuscript. In general, the English and paper format is quite poor, for example, from line 235 to line 240, Table 5.

Reviewers are right, changes have been made to all text.

  1. What are the significant findings of the manuscript? The author’s team should clarify the featured results of this research.

It was inserted as the last topic of the conclusion: In general, cementitious paste mixtures with the addition of SAP improve the properties analyzed. However, due to the results found for mixture 6, future works can be studied for other curing processes.

  1. The content of the Introduction should be revised into a condensed version. Some information can be shifted to the following parts.

       Reviewers are right, changes have been made to all text.

  1. The research objectives and the significance of the research should be clarified in the manuscript.

Reviewers are correct; we updated the obectives to make it more clear.

  1. There is a lack of citations for the reviews and the literature content should be improved.

Reviewers are correct; we updated the literature.

  1. The material properties used in this research should be clarified, especially the cement properties, and pozzolanic properties.

       Cement  and pozzolanic foram realizados: massa específica conforme (ABNT NBR 16605:2017)  e granulometria a laser, que ainda não possui norma específica.

  1. The curing process of the specimens is not specified. The replicates for engineering tests were not shown.

      Information inserted before table 2, about replicates showed table 2.

  1. How to determine the 600μm and 800μm particle sizes?

       For the superabsorbent polymer, the specifications are Specific mass (g/cm³)=1,1 and Absorption (g of water/g of SAP)= 200-400 according to IGTPAN, 2022.

  1. Why did the author’s team conduct the mini-slump test instead of the conventional slump test? It should be specified. The slump test results should be presented in Figure format.

      For self-compacting paste this is the conventional slump test. Unfortunately, the rehearsal photos were not taken.

  1. The unit of the unconfined compressive strength test was not shown.

       The unit (MPa) was inserted in the legend of the figure 2.

  1. The author’s team should clarify why the mixture having CPV0.15% 800μm dropped significantly. Meanwhile, with the same cement type, the 600μm mixture obtained the highest compressive strength.

      The authors improved and highlighted the 2nd paragraph on page 10 with the requested explanation:

However, BALOCH et al. (2019) [9], state that the SAP can exhibit two behaviors: it will enhance hydration processes, assisting in the internal curing process and boosting strength, or it can induce voids in the matrix, weakening strength. This last theory ex-plains the performance of mixture no. 6 in the axial compression test is that the inclusion of SAP in this mixture may have contributed to the second scenario reported, where the mechanical compression strength was decreased by creating voids in the cement matrix. In the study indicated above, there were two different methods of curing used: in water and in air. The mixes treated in water revealed matrix holes and had decreased strength, whereas the samples cured in air had increased strength.

Again, I appreciate your availability for reviewing and indicating improvements in the work. If you have any questions or other suggestions, please do not hesitate to contact me.

Sincerely,

Patricia Capellato

Round 2

Reviewer 1 Report

The authors have addressed my questions.

Author Response

13th November 2022

The authors of the manuscript and I are very grateful for your review and considerations. Rest assured that all of your observations were carefully analyzed and they undoubtedly enriched the result of the work. Thus, you review English language and style.

Again, I appreciate your availability for reviewing and indicating improvements in the work. If you have any questions or other suggestions, please do not hesitate to contact me.

Sincerely,

Patricia Capellato

Reviewer 3 Report

The paper was improved and the content is acceptable

Author Response

(The authors gave the same response as above.)
